# A conserved ubiquitin- and ESCRT-dependent pathway internalizes human lysosomal membrane proteins for degradation

**Weichao Zhang**[1], **Xi Yang**[1], **Liang Chen**[1], **Yun-Yu Liu**[1], **Varsha Venkatarangan**[1], **Lucas Reist**[1], **Phyllis Hanson**[2], **Haoxing Xu**[1], **Yanzhuang Wang**[1], **Ming Li**[1]*

**1** Department of Molecular, Cellular, and Developmental Biology, University of Michigan, Ann Arbor, Michigan, United States of America, **2** Department of Biological Chemistry, University of Michigan, Ann Arbor, Michigan, United States of America

* mlium@umich.edu

**Data Availability Statement:** The flow cytometry data are uploaded to the Flow Repository (ID#s: FR-FCM-Z45X, M, N, L, K). Remaining relevant

## Abstract

The lysosome is an essential organelle to recycle cellular materials and maintain nutrient homeostasis, but the mechanism to down-regulate its membrane proteins is poorly understood. In this study, we performed a cycloheximide (CHX) chase assay to measure the half-lives of approximately 30 human lysosomal membrane proteins (LMPs) and identified RNF152 and LAPTM4A as short-lived membrane proteins. The degradation of both proteins is ubiquitin dependent. RNF152 is a transmembrane E3 ligase that ubiquitinates itself, whereas LAPTM4A uses its carboxyl-terminal PY motifs to recruit NEDD4-1 for ubiquitination. After ubiquitination, they are internalized into the lysosome lumen by the endosomal sorting complexes required for transport (ESCRT) machinery for degradation. Strikingly, when ectopically expressed in budding yeast, human RNF152 is still degraded by the vacuole (yeast lysosome) in an ESCRT-dependent manner. Thus, our study uncovered a conserved mechanism to down-regulate lysosome membrane proteins.

## Introduction

As an essential organelle, the lysosome is responsible for various cellular processes, including protein turnover and recycling, energy metabolism, intracellular signaling, and nutrient storage [1–3]. The lysosome membrane contains hundreds of transmembrane proteins, many of which are transporters and channels that shuttle metabolites (ions, amino acids, cholesterol, etc.) across the membrane [4–7]. Malfunction of these lysosomal membrane proteins (LMPs) can give rise to inherited genetic disorders called lysosomal storage diseases (LSDs). Many LSD patients will develop severe neurodegeneration symptoms [8]. Furthermore, growing evidence suggests that mutations in LMPs and other lysosome dysfunction are associated with age-related neurodegeneration such as Alzheimer disease, frontotemporal dementia, and Parkinson disease [9–11]. As we age, the lysosome membrane gradually accumulates damaged proteins and loses its integrity, which dampens the cell's ability to remove pathogenic protein aggregates and damaged organelles, eventually leading to cell death and inflammation [12–15].

data are within the paper and its Supporting Information files.

**Funding:** This project has been funded by grants from National Institutes of Health, with R01GM133873 to ML, R35GM130331 to YW, and R01GM122434 to PH. The funders had no role in study design, data collection and analysis, decision to publish, or preparation of the manuscript.

**Competing interests:** The authors have declared that no competing interests exist.

**Abbreviations:** CHX, cycloheximide; CMA, chaperone-mediated autophagy; CTSD, cathepsin D; DMEM, Dulbecco's Modified Eagle Medium; ERAD, endoplasmic reticulum–associated degradation; ESCRT, endosomal sorting complexes required for transport; FACS, fluorescence-activated cell sorting; Gb3, globotriaosylceramide; ILV, intralumenal vesicle; LAMP2, lysosome-associated membrane protein 2; LLOMe, L-Leucyl-L-leucine methyl ester; LSD, lysosomal storage disease; LMP, lysosomal membrane protein; MOI, multiplicity of infection; sgRNA, single-guide RNA; shRNA, short hairpin RNA; siRNA, small interfering RNA; STED, Stimulated Emission Depletion; UPS, ubiquitin-proteasome system; WT, wild-type; YNB, Yeast Nitrogen Base.

Strategies to maintain the lysosome membrane integrity during aging will likely delay the onset of neurodegenerative symptoms.

Given the physiological importance and clinical implications of LMPs, we wonder how human LMPs are regulated and quality controlled. At the organelle level, if the lysosomal membrane is mildly damaged by insults like lysosomotropic compounds, such as L-leucyl-L-leucine methyl ester (LLOMe) or iron-dependent oxidative stress, the endosomal sorting complexes required for transport (ESCRT) machinery can be recruited to the lysosome surface to repair the membrane [16–18]. If the damage is too severe to be repaired, ruptured lysosomes will be sequestered and degraded by selective autophagy, a process termed "lysophagy" [19,20].

At the protein level, selective removal of proteins from the lysosome surface is essential for adjusting membrane composition in response to environmental cues. However, very little is known about its underlying mechanism. A process like lysophagy, which engulfs whole lysosomes, could not accomplish selectivity. This leads to important questions as to how human lysosomes selectively down-regulate their membrane proteins and what machinery might be involved in the process.

In this study, we screened approximately 30 human LMPs using a cycloheximide (CHX) chase assay and identified a few candidates with short half-lives. Among those candidates, we focused on RNF152 (a lysosome membrane–anchored E3 ligase) and LAPTM4A (a 4-transmembrane LMP) as cargoes to examine the possible mechanisms of LMP turnover. We discovered that their degradation is both ubiquitination- and lysosome dependent. Further, we showed that the conserved ESCRT machinery plays a vital role in cargo internalization. Collectively, our work suggests that the ubiquitin- and ESCRT-dependent degradation pathway is a conserved and general mechanism to down-regulate LMPs.

## Results

### Cycloheximide chase screen to determine the half-lives of LMPs

To identify fast-degrading lysosome membrane substrates, we collected more than 30 GFP- or mCherry-tagged human lysosome membrane proteins. These proteins were identified as lysosomal by either microscopy or mass spectrometry studies [4–7,21]. We transiently expressed them in HEK293 cells and determined their half-lives using a CHX chase assay [22]. Here, we present some examples of the screen (Fig 1A and 1B, S1 Fig, S1B Table). While most LMPs are very stable (TMEM175, PQLC2, CTNS, CLCN7, lysosome-associated membrane protein 2 [LAMP2], and TMEM192), a few LMPs have a relatively short half-life (LAPTM4A $t_{1/2}$ = 11.0 hours, RNF152 $t_{1/2}$ = 6.7 hours, and OCA2 $t_{1/2}$ = 5.6 hours; Fig 1A and 1B). For LAPTM4A, RNF152, and OCA2, we confirmed their lysosome localization by co-staining with LAMP2 (Fig 1C).

Although OCA2 has the fastest degradation, it is a melanosome protein absent in normal lysosomes [23]. Studying the degradation of OCA2 in cells other than melanocytes may be physiologically irrelevant. Therefore, we focused on RNF152 and LAPTM4A to investigate the degradation pathway of lysosome membrane proteins.

### RNF152 is a protein of short half-life

RNF152 is a single pass, type II transmembrane protein with 203 amino acids (Fig 2A). The predicted molecular mass is 23 kDa. It was recently characterized as a lysosome E3 ligase that negatively regulates mTORC1 activity by ubiquitinating RagA and Rheb [24,25]. We generated an antibody to confirm that the overexpressed non-tagged RNF152 is also degraded (Fig 2B). However, this antibody failed to detect the endogenous RNF152 (S2A Fig). After slightly elevating the RNF152 level using a leaky TET-ON promoter, we were able to show that the weakly

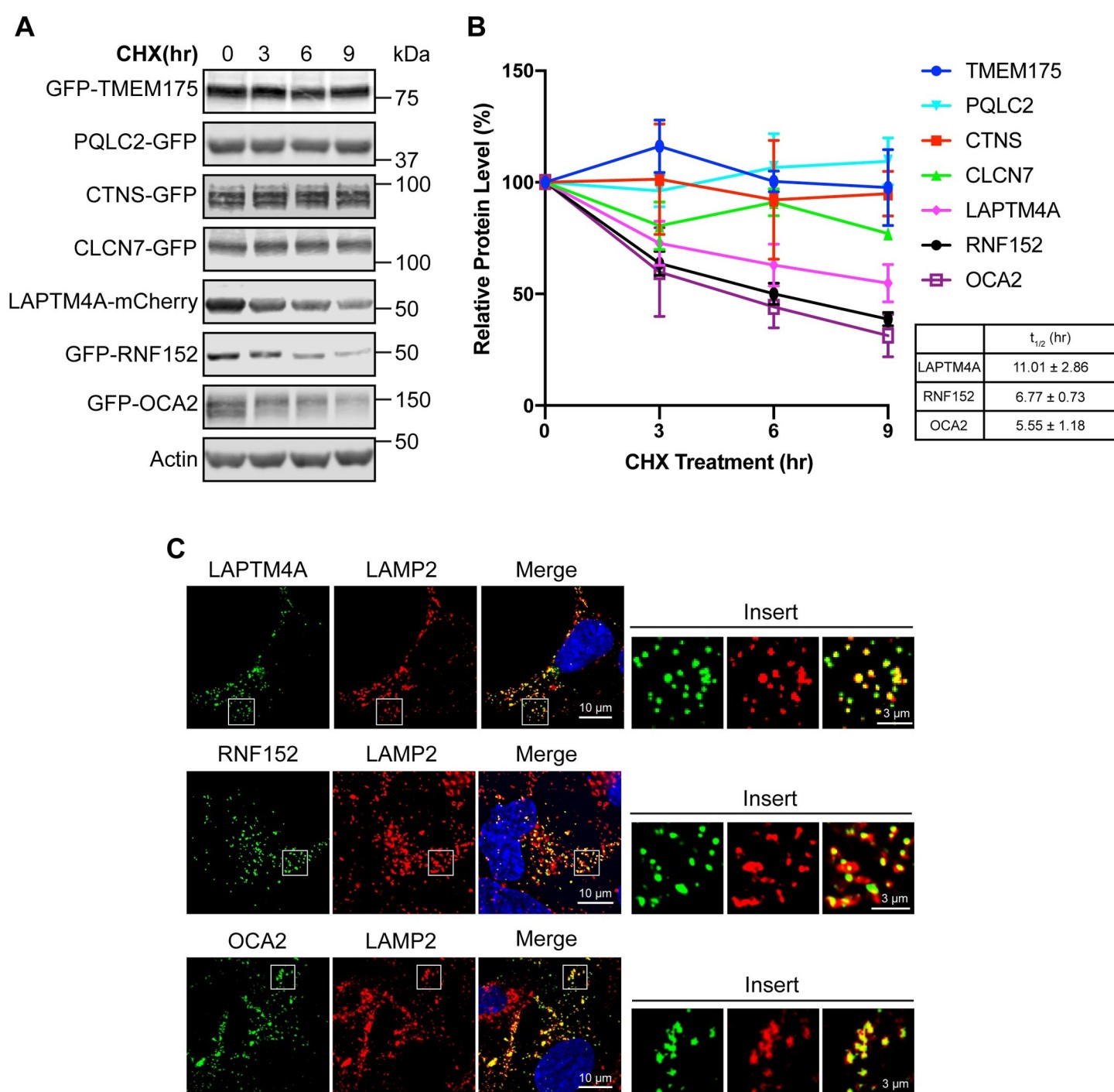

**Fig 1. CHX chase assay to determine the stability of LMPs. (A)** CHX chase assay to determine the stability of transiently expressed LMPs in HEK293 cells. **(B)** Quantification of the protein levels in A, $n$ = 3. Error bars represent standard deviation. The half-lives ($t_{1/2}$) of LAPTM4A, RNF152, and OCA2 were listed in the table. **(C)** Immunofluorescence showing the localization of LAPTM4A-mCherry, GFP-RNF152, and GFP-OCA2 in HEK293 cells co-stained with LAMP2. Uncropped western blots and microscopy images are provided in S1 Raw Images. The numerical data underlying the plot can be found in S1 Data. CHX, cycloheximide; LAMP2, lysosome-associated membrane protein 2; LMP, lysosomal membrane protein.

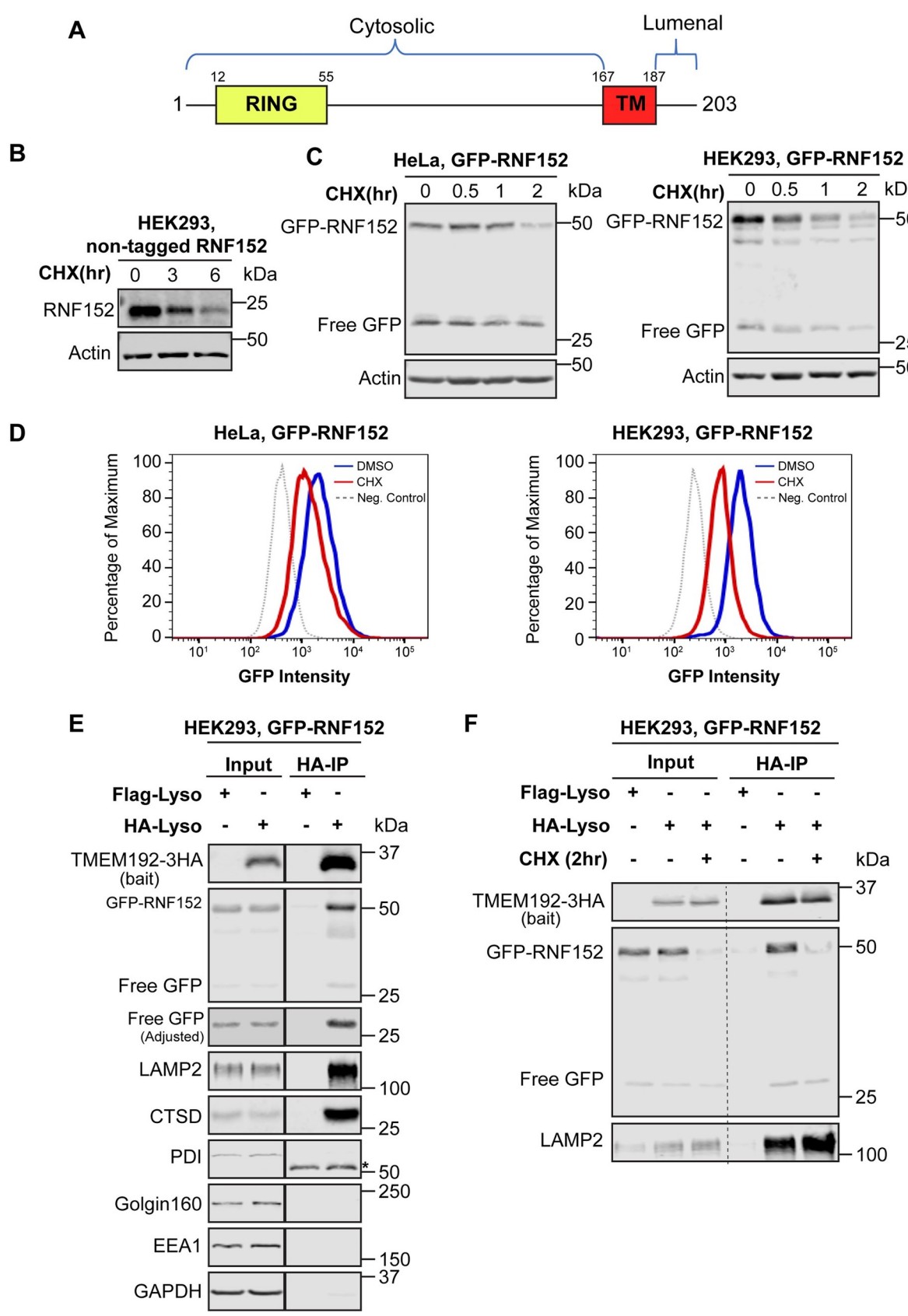

**Fig 2. RNF152 is a fast-degrading LMP. (A)** A schematic representation of RNF152. **(B)** CHX chase assay of transiently expressed, non-tagged RNF152 in HEK293 cells. **(C)** CHX chase assay of stably expressed GFP-RNF152 in both HeLa and HEK293 cells. **(D)** Flow cytometry analysis of stably expressed GFP-RNF152. Cells were pretreated with either CHX or DMSO for 2 hours. Noninfected cells served as negative controls. **(E)** Both full-length GFP-RNF152 (50 kDa) and free GFP (27 kDa) were enriched by Lyso-IP. Flag-Lyso: cells stably expressing TMEM192-2XFLAG; HA-Lyso: cells stable expressing TMEM192-3HA. Both cells were stably expressing GFP-RNF152. Asterisk: mouse heavy chain. **(F)** Lysosome-localized GFP-RNF152 was degraded after 2 hours of CHX treatment. LAMP2 serves as an internal control. Uncropped western blots are provided in S1 Raw Images. CHX, cycloheximide; LAMP2, lysosome-associated membrane protein 2; LMP, lysosomal membrane protein.

expressed RNF152 is quickly degraded (S2B Fig). The BafA1 treatment can stabilize both endogenous and leaky expression RNF152 (S2A and S2B Fig). For further characterizing the mechanism of RNF152 degradation, we used the overexpressed GFP-RNF152. This fusion protein is still functional as it can interact and ubiquitinate RagA (S2C Fig).

The initial screen was conducted with transient transfection, which had a high variation of expression levels from cell to cell. Thus, we generated stable cell lines (HEK293 and HeLa) expressing GFP-RNF152 by lentivirus transduction. Repeating the CHX chase assay with these stable lines indicated that GFP-RNF152 is mostly degraded within 2 hours (Fig 2C), which was much faster than the transient system. Besides western blots, we also used flow cytometry to confirm the fast degradation of GFP-RNF152 in both cell lines (Fig 2D).

Interestingly, we observed a 27-kDa band that is about the size of free GFP in the CHX chase assay (Fig 2C). Because GFP is fused to the cytosolic side of RNF152 (Fig 2A), there are 2 possibilities to generate the 27-kDa band. First, GFP-RNF152 may be internalized into the lysosome, and free GFP is the by-product of degradation. Second, free GFP may be cleaved off RNF152 due to the instability of the fusion protein. In this case, free GFP would be in the cytosol. To differentiate between these 2 possibilities, we adopted the Lyso-IP experiment developed by the Abu-Remaileh and colleagues [26]. It uses TMEM192-3HA, a lysosome-specific membrane protein, as the bait to pull down intact lysosomes. As expected, full-length GFP-RNF152 and other lysosome proteins such as LAMP2 and cathepsin D (CTSD) were enriched by Lyso-IP. In contrast, PDI (ER), Golgin160 (Golgi), EEA1 (endosomes), and GAPDH (cytosol) were absent (Fig 2E). Importantly, we found that the 27-kDa band was also enriched by Lyso-IP, indicating that free GFP is inside the lumen. Thus, RNF152 may be internalized and degraded in the lysosome.

Lastly, a small fraction of RNF152 was also reported to localize to endosomes [27]. To ensure that the lysosome population of RNF152 is fast degrading, we performed the Lyso-IP experiment using cells collected before and after 2 hours of CHX treatment. Our result confirmed that the lysosome population of GFP-RNF152 is rapidly turned over, while LAMP2 is very stable (Fig 2F).

## RNF152 degradation is ubiquitination dependent

Protein ubiquitination is the prerequisite for degradation in eukaryotic cells. Given that RNF152 has a short half-life, we hypothesized that RNF152 might be ubiquitinated. The full-length GFP-RNF152 migrates at 50 kDa. Interestingly, we observed a faint 60-kDa band that might be the monoubiquitinated form (Fig 3A, left panel). After longer exposure, a high molecular mass smear appeared on top of the full-length protein (Fig 3A, right panel). Could this smear be the polyubiquitinated GFP-RNF152? To answer this, we transfected HA-tagged ubiquitin into cells that stably express either GFP-RNF152 or GFP control, followed by immunoprecipitation using the GFP-TRAP nanobody. HA-ubiquitin was incorporated into the high molecular mass smear of GFP-RNF152, but not GFP control (Fig 3B). This result confirmed that a significant portion of GFP-RNF152 is polyubiquitinated.

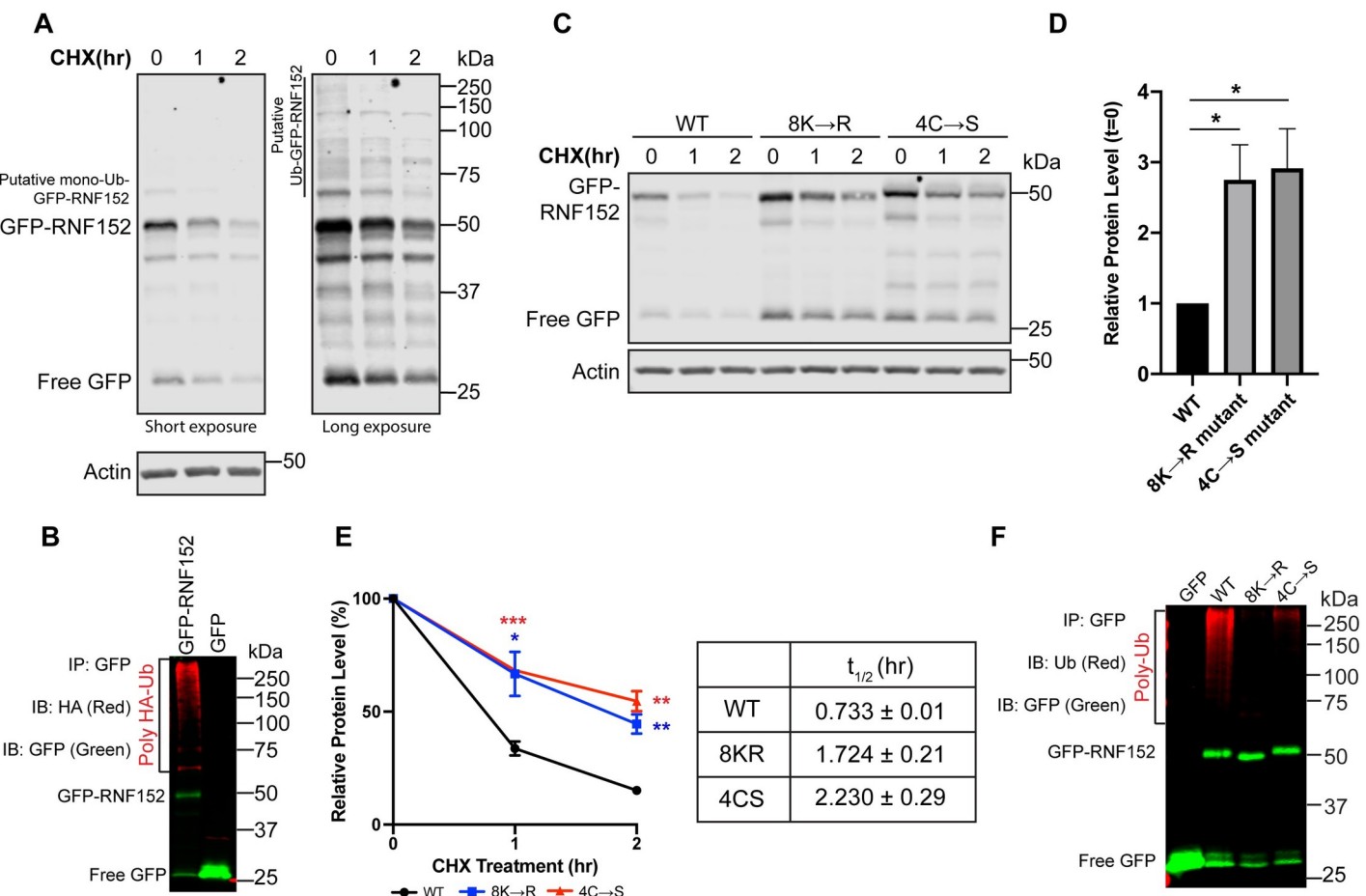

**Fig 3. RNF152 degradation is ubiquitin dependent.** (A) Left: CHX chase assay of stably expressed GFP-RNF152. Right: long exposure. (B) GFP-RNF152, but not free GFP, is polyubiquitinated. (C) CHX chase assay of stably expressed WT, 8K→R, and 4C→S mutants of GFP-RNF152. (D) Steady-state (0 hour) full-length protein levels in C, $n$ = 3. Error bars represent standard deviation. * $p \leq 0.05$. (E) Left: quantification of GFP-RNF152 degradation in C, $n$ = 3. Error bars represent standard deviation. * $p \leq 0.05$, ** $p \leq 0.01$, *** $p \leq 0.001$. Table: Calculated protein half-lives. (F) 8K→R and 4C→S mutants have impaired polyubiquitination. Uncropped western blots are provided in S1 Raw Images. The numerical data underlying the plot can be found in S1 Data. CHX, cycloheximide; WT, wild-type.

To test if polyubiquitination is essential for the degradation, we mutated all 8 lysines in the cytosolic domain of RNF152 to arginine (8K→R). As shown in Fig 3C and 3D, the 8K→R mutation increased the steady-state level of GFP-RNF152$_{8K \rightarrow R}$ (2.7 fold), and its degradation kinetics was significantly slower (Fig 3E).

Because RNF152 is a RING domain E3 ligase, we hypothesized that its degradation might be due to autoubiquitination. To test this, we mutated 4 cysteines of the RING finger motif to serines (4C→S mutant) [25]. Similar to the 8K→R mutant, the steady-state protein level of GFP-RNF152$_{4C \rightarrow S}$ was increased by 2.8-fold (Fig 3C and 3D), and the degradation kinetics was also significantly slower (Fig 3E). Lastly, the ubiquitin blots confirmed that both 8K→R and 4C→S mutants have reduced ubiquitination (Fig 3F).

Taken together, we concluded that the degradation of GFP-RNF152 is a ubiquitin-dependent process. The short half-life of GFP-RNF152 is likely due to autoubiquitination.

## RNF152 is mainly degraded by the lysosome

In eukaryotic cells, there are 2 primary mechanisms to break down proteins: proteasome-dependent and lysosome-dependent proteolysis. The partial accumulation of GFP inside the

lysosome suggests that RNF152 is degraded by the lysosome. To test this further, we treated cells with either lysosome V-ATPase inhibitor Bafilomycin A1 (BafA1) or proteasome inhibitor MG132. BafA1 treatment results in a rapid neutralization of lysosome pH and inhibition of the lumenal proteases. After 6 hours of BafA1 treatment, the steady-state GFP-RNF152 level increased 2-fold (Fig 4A and 4B). In contrast, MG132 treatment did not lead to a significant change in protein levels. The double treatment did not further increase the steady-state RNF152 than BafA1 alone.

We also observed an accumulation of a high molecular mass smear in the BafA1-treated sample and the double treatment sample (Fig 4A and 4C). Co-expressing HA-ubiquitin with GFP-RNF152 verified that they were polyubiquitinated (Fig 4D). These results suggest that the lysosome, but not proteasome, is the primary location to degrade RNF152.

We further confirmed our findings with both flow cytometry and microscopy. As shown in Fig 4E, the GFP intensity only increased slightly after MG132 treatment. In contrast, BafA1 treatment led to a major climb in GFP intensity. The BafA1+MG132 double treatment did not further increase the intensity. Consistently, under a microscope, we found the GFP signal was also much higher after BafA1 treatment (Fig 4F). Most of the GFP signal colocalized with LAMP2. Importantly, after deconvolving the imaging results (SoftWorx, GE Healthcare Life Sciences, Chicago, Illinois, USA), we observed that much of the GFP signal was inside the lumen (line scan in Fig 4G). Similar results were also obtained with a super-resolution Leica Stimulated Emission Depletion (STED) microscope (Leica Microsystems, Morrisville, North Carolina, USA) (S3 Fig). The lumenal GFP strongly supports a model that RNF152 is internalized into the lysosome and degraded by the lumenal proteases.

Besides the steady-state protein levels, we also compared the degradation kinetics of GFP-RNF152. As shown in S4 Fig, the degradation of GFP-RNF152 was blocked by BafA1 treatment, even after we extended the chase to 6 hours. In contrast, MG132 only delays the degradation of GFP-RNF152. Interestingly, we observed an increase of free GFP after both MG132 and BafA1 treatment. MG132 is known to partially inhibit some lysosome proteases such as cathepsin A and cathepsin B [28,29]. The delayed degradation of GFP-RNF152 and increased free GFP signal in MG132 treated samples might be due to the reduced activity of some lysosome proteases.

Taken together, we concluded that RNF152 is degraded through lysosome-dependent proteolysis. For subsequent studies, we focused on identifying the machinery that delivers RNF152 into the lumen.

## Macroautophagy machinery and CMA pathway are not involved in the degradation of GFP-RNF152

In mammalian cells, there are 4 possible mechanisms to deliver intracellular materials into lysosomes for degradation: macroautophagy, microautophagy, ESCRT-dependent formation of intraluminal vesicles, and chaperone-mediated autophagy (CMA) [30]. How is ubiquitinated RNF152 internalized into the lysosome then? Although lysophagy can deliver an entire damaged lysosome into other healthy lysosomes for degradation [31], it cannot selectively turnover a particular membrane protein while leaving others intact. Recently, Lee and colleagues reported that glucose starvation and certain drug treatments could trigger a microautophagy process to selectively turnover some LMPs. Although the mechanism remains to be identified, it was shown that the LC3 lipidation machinery, such as ATG5, is critical to initiate microautophagy [32]. To test if this microautophagy is involved in RNF152 degradation, we knocked out either ATG5 or ATG7 using the CRISPR/Cas-9 method [33,34]. In wild-type (WT) cells, Atg5 forms a stable 55-kDa conjugate with Atg12 in an Atg7-dependent manner

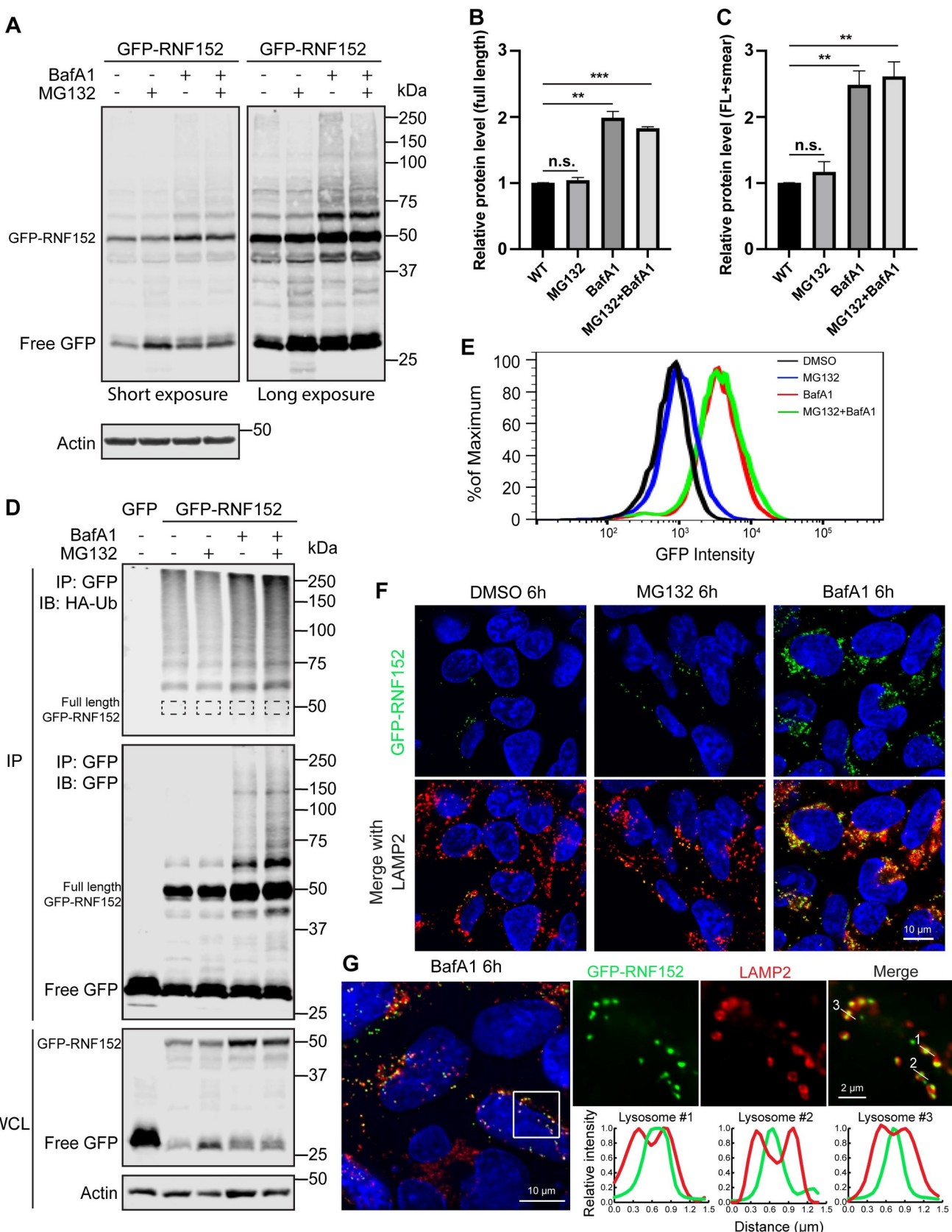

**Fig 4. RNF152 is degraded in the lysosome. (A)** Left: GFP-RNF152 protein levels after MG132 (50 μM), BafA1 (400 nM), and double treatment for 6 hours. Right: long exposure. **(B)** Quantification of the full-length GFP-RNF152 protein level in A, n = 3. Error bars represent standard deviation. n.s., not significant. ** $p \leq 0.01$, *** $p \leq 0.001$. **(C)** Quantification of the total GFP-RNF152 protein level, including full-length and the high molecular mass smear in A, n = 3. **(D)** Accumulation of polyubiquitinated GFP-RNF152 after the BafA1 and BafA1+MG132 treatments. **(E)** Flow cytometry analysis of stably express GFP-RNF152 in HEK293 cells after 6 hours of indicated treatments. **(F)** Localization of GFP-RNF152 and LAMP2 after indicated treatments. Images were taken by a confocal microscope. **(G)** Localization of GFP-RNF152 and LAMP2 after BafA1 (400 nM) treatment. The image was taken by a DeltaVision deconvolution microscope. Line scan analysis highlighted the lumenal localization of GFP-RNF152. Uncropped western blots and microscopy images are provided in S1 Raw Images. The numerical data underlying the plot can be found in S1 Data. WT, wild-type.

(S5A Fig, left 3 lanes) [35]. After knocking out Atg7, the conjugate no longer forms, and Atg5 appears as a 33-kDa band (S5A Fig, last 3 lanes). However, neither ATG5 nor ATG7 knockout cells exhibited any defect in GFP-RNF152 degradation (S5A and S5B Fig).

Autophagy and the ubiquitin-proteasome system (UPS) are the 2 major pathways to degrade proteins in eukaryotic cells [36]. There is mounting evidence to show that the 2 pathways can crosstalk. It is possible that the UPS pathway is up-regulated to compensate for the loss of autophagy [37–40]. To rule out the possibility that GFP-RNF152 is redirected to the proteasome after knocking out macroautophagy, we treated the ATG7KO cells with BafA1 and MG132. As shown in S5C and S5D Fig, GFP-RNF152 degradation is still mainly dependent on the lysosome in autophagy-deficient cells. Thus, the macroautophagy machinery and likely the LC3 lipidation–triggered microautophagy is not involved in RNF152 degradation.

We also examined whether CMA is involved. In the CMA pathway, the chaperone Hsc70 recognizes a KFERQ-like motif of its substrates and delivers them to the lysosome for degradation [41]. Using a web-based motif finder, we identified one putative KFERQ-like motif in the cytosolic domain of RNF152: $^{46}$QKDVR$^{50}$ (S5E Fig) [42]. However, mutating $^{46}$QK$^{47}$ to AA does not affect RNF152 degradation (S5F and S5G Fig), suggesting that the CMA pathway may not be involved.

## The early-stage ESCRTs are less critical for GFP-RNF152 degradation

Next, we asked if ESCRT machinery is responsible for internalizing ubiquitinated GFP-RNF152. The ESCRT machinery is composed of several subcomplexes, including ESCRT-0, I, II, III, and the AAA-ATPase VPS4, all of which are conserved from yeast to human [43]. ESCRTs usually function on the endosome surface to sort ubiquitinated cargoes into the lumen as intralumenal vesicles (ILVs) [44]. Besides the ILV formation, mammalian ESCRTs play diverse roles at different membrane compartments. Recently, it was demonstrated that the ESCRT machinery is involved in the repair of damaged lysosome membranes [17,18]. However, whether the ESCRT machinery can internalize ubiquitinated lysosome membrane proteins remains an open question. To answer this, we knocked down different components of the ESCRT machinery and tested if they affect RNF152 degradation.

We first knocked down TSG101 (ESCRT-I, Vps23 in yeast) and ALIX (accessory unit, Bro1 in yeast) using small interfering RNA (siRNA) because they are required to recruit ESCRT-III for lysosome membrane repair [17,18]. Knocking down ALIX alone did not affect RNF152 degradation. However, knocking down either TSG101 or both TSG101 and ALIX had a slight impact on the degradation kinetics (S6A and S6B Fig). There was also a marginal increase of the ubiquitinated GFP-RNF152 in TSG101 knockdown and TSG101+ALIX double knockdown groups (S6A Fig).

We then tested ESCRT-0 and ESCRT-II by knocking down HRS (Vps27 in yeast) and VPS22, respectively. Both knockdowns had a minor impact on the degradation of GFP-RNF152 (S6C and S6D Fig). In the case of VPS22, there was a slight rise of the ubiquitinated GFP-RNF152 (S6C Fig).

To ensure that the siRNA knockdown methods can disrupt the function of ESCRT machinery, we stained the cells with the FK2 antibody that specifically recognizes polyubiquitin [45].

Since ESCRT machinery is responsible for internalizing ubiquitinated proteins from the endosome membrane, depleting ESCRTs will lead to the accumulation of polyubiquitinated proteins at the endosome surface. Indeed, after knocking down either TSG101+ALIX or Vps22, we observed enlarged vacuole-like structures that stained positive for FK2 antibody (S6E Fig). In HRS knockdown, we did not observe the accumulation of polyubiquitin signals (S6E Fig) even though the western blot showed a pronounced reduction of the HRS level (S6C Fig).

Taken together, we concluded that early ESCRT components, especially TSG101, might play a role in RNF152 degradation. However, the relatively weak phenotypes suggested other redundant components might be involved to recruit the ESCRT-III.

## The late-stage ESCRTs are important for GFP-RNF152 degradation

We then tested the ESCRT-III components, which polymerize into spiral filaments to drive the internalization of ubiquitinated membrane cargoes. After knocking down both CHMP4A and CHMP4B (Snf7 in yeast), the degradation of GFP-RNF152 is significantly delayed, as demonstrated by both western blots (Fig 5A and 5B) and flow cytometry (Fig 5C). Besides, we observed a substantial accumulation of ubiquitinated GFP-RNF152 (Fig 5A). We also used imaging to check the internalization of RNF152. In the siRNA control, GFP-RNF152 accumulated in the lumen after BafA1 treatment. However, it was stabilized on the membrane of vacuole-like structures after the double knockdown of CHMP4A and CHMP4B (Fig 5D and 5E). Together, these results indicated that ESCRT-III plays a critical role in the degradation of GFP-RNF152.

Besides ESCRT-III, we also tested the AAA-ATPase VPS4 responsible for the disassembly of ESCRT-III filaments [46]. Here, we used an inducible TET-ON system to overexpress either HA-VPS4A or its dominant-negative mutant E228Q [47]. Overexpression of both WT HA-VPS4A and the E228Q mutant delayed the degradation of GFP-RNF152, with the EQ mutant having a more potent effect (Fig 5F and 5G). Consistent results were also observed by flow cytometry analysis (Fig 5H). Furthermore, for both WT and E228Q mutant, we observed the accumulation of polyubiquitinated GFP-RNF152 (Fig 5F). These results demonstrated that (1) VPS4 is important for RNF152 degradation; and (2) overexpression of WT HA-VPS4A also partially disrupts the ESCRT function, likely due to its HA tag.

To further confirm the importance of VPS4, we stained the cells with the FK2 antibody that recognizes polyubiquitin. Overexpression of both WT and the E228Q mutant showed an accumulation of FK2 positive signals, with the mutant having a much stronger effect (S7 Fig). Specifically, after overexpressing WT HA-VPS4A, approximately 50% of cells displayed FK2 positive signals on enlarged vacuole-like structures. In contrast, with the E228Q mutant, more than 90% of cells were stained positive for FK2 antibody, and vacuoles were much more prominent. These results were consistent with the analysis using GFP-RNF152 in Fig 5F–5H.

Taken together, our results strongly suggest that the late-stage ESCRT machinery, specifically ESCRT-III and VPS4, is critical for the degradation of GFP-RNF152.

## Degradation of LAPTM4A depends on NEDD4-1 and the ESCRT machinery

To test if the ubiquitin- and ESCRT-dependent LMP degradation is a general mechanism, we characterized the turnover of LAPTM4A. It contains 4 transmembrane helices, with both N-terminal and carboxyl terminus facing the cytoplasm (Fig 6A). Recent studies revealed that LAPTM4A is required for globotriaosylceramide (Gb3) synthesis [48,49]. However, its localization remains controversial. One study reported that overexpressed LAPTM4A localized to Golgi [48], which is inconsistent with our observation (Fig 1C). We first confirmed that

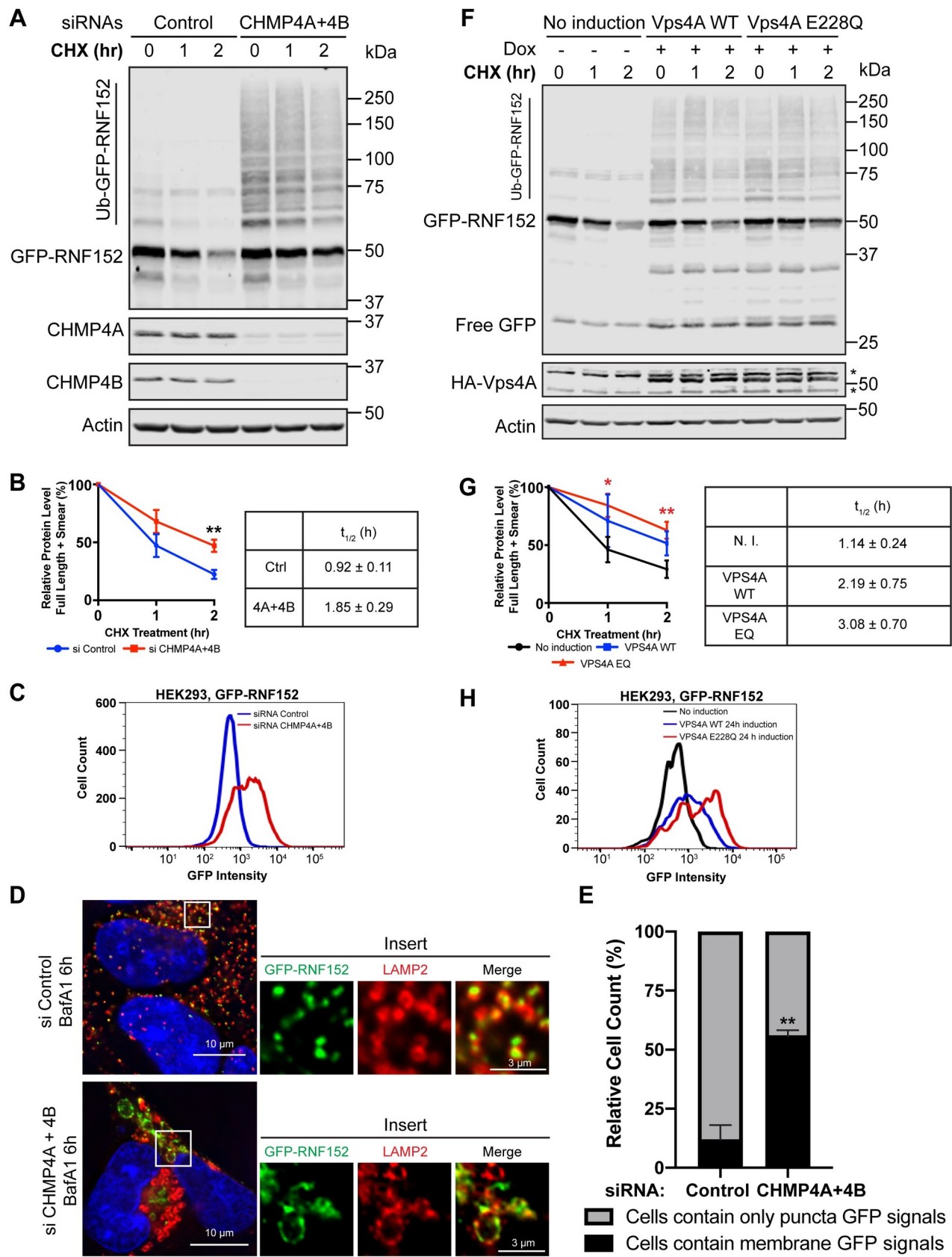

**Fig 5. ESCRT-III and VPS4 are important for the degradation of RNF152. (A)** CHX chase assay of GFP-RNF152 in siRNA control and CHMP4A + CHMP4B double knockdown cells. **(B)** Left: total GFP-RNF152 protein levels in A, $n$ = 3. Error bars represent standard deviation. $^{**}$ $p \leq 0.01$. Right: half-lives of GFP-RNF152 in A. **(C)** Flow cytometry analysis of GFP-RNF152 in siRNA control and CHMP4A + CHMP4B double knockdown cells. **(D)** Localization of GFP-RNF152 and LAMP2 in siRNA control and CHMP4A +CHMP4B double knockdown cells after BafA1 (400 nM) treatment. **(E)** Quantification of D counting the percentage of cells that contain either punctate or membrane GFP signals. **(F)** CHX chase assay of GFP-RNF152 in indicated cells. **(G)** Left: total GFP-RNF152 protein levels in G, $n$ = 3. Error bars represent standard deviation. $^{*}$ $p \leq 0.05$, $^{**}$ $p \leq 0.01$. Right: half-lives of GFP-RNF152 in F. **(H)** Flow cytometry analysis of GFP-RNF152 in indicated cells. Uncropped western blots and microscopy images are provided in S1 Raw Images. The numerical data underlying the plot can be found in S1 Data. CHX, cycloheximide; ESCRT, endosomal sorting complexes required for transport; siRNA, small interfering RNA; WT, wild-type.

endogenous LAPTM4A is lysosome localized and quickly degraded (Fig 6B and 6C). As shown in Fig 6B, the endogenous LAPTM4A was co-purified with lysosomes, whereas the Golgi marker Golgin 160 was absent. Besides the 22-kDa full-length LAPTM4A, we also observed a 17-kDa truncation product (asterisk, Fig 6B). Both bands were rapidly degraded after adding CHX (Fig 6C).

In addition to the endogenous protein, we also characterized stably expressed LAPT-M4A-GFP. Consistently, LAPTM4A-GFP appeared as 2 major bands (50 and 45 kDa), and both bands decreased after CHX treatment (Fig 6D). Pretreating cells with BafA1, but not MG132, stabilized the protein by more than 2-fold (Fig 6E and 6F). Besides, there was an accumulation of a high molecular mass smear in BafA1 treated samples (Fig 6F), which was confirmed as the polyubiquitinated LAPTM4A-GFP (Fig 6G). But what E3 ligase might be responsible for its ubiquitination?

Examining the protein sequence revealed that LAPTM4A contains 3 conserved PY motifs at its carboxyl terminus (PPxY or LPxY, where x can be any amino acid, Fig 6A). The PY motif has been implicated in recruiting the NEDD4 family E3 ligases to various membrane compartments [50–54]. We confirmed that LAPTM4A does interact with NEDD4-1, which depends on its PY motifs (Fig 6H). Following this clue, we either mutated all 3 PY motifs or knocked down NEDD4-1. In both experiments, LAPTM4A was stabilized (Fig 6I and 6J). Together, they support a model where NEDD4-1 is recruited to the lysosome membrane to ubiquitinate LAPTM4A by interacting with its PY motifs (Fig 6K).

We then tested if the ESCRT machinery is essential for LAPTM4A degradation. Like RNF152, knocking down HRS and VPS22 had either no effect (HRS) or only a minor effect (VPS22) on the degradation. In contrast, the double silencing of both TSG101 and ALIX had a more substantial impact, and an accumulation of the ubiquitinated LAPTM4A was observed (Fig 7A and 7B). Knocking down the ESCRT-III (CHMP4A+CHMP4B) had an even stronger impact on LAPTM4A degradation and the accumulation of polyubiquitinated forms (Fig 7C and 7D). Similar strong effects were also observed when we overexpressed the dominant-negative E228Q mutant of VPS4A (Fig 7E and 7F). Lastly, we confirmed that endogenous LAPTM4A degradation also depends on late ESCRT, such as CHMP4A and CHMP4B (Fig 7G and 7H).

In summary, our analysis of LAPTM4A further supports that the ubiquitin- and ESCRT-dependent pathway is a general mechanism to degrade LMPs.

## A conserved pathway from yeast to human

Fundamental biological processes are generally conserved. Here, our study uncovered a ubiquitin- and ESCRT- dependent pathway in humans to degrade lysosome membrane proteins. Is this pathway conserved in yeast? To directly answer the question, we expressed GFP-RNF152 in budding yeast. To our satisfaction, GFP-RNF152 is still correctly localized to the vacuole membrane (Fig 8). Strikingly, when protein synthesis was stopped by CHX, GFP-RNF152 was quickly internalized into the lumen and degraded (Fig 8A and 8B). Deletion

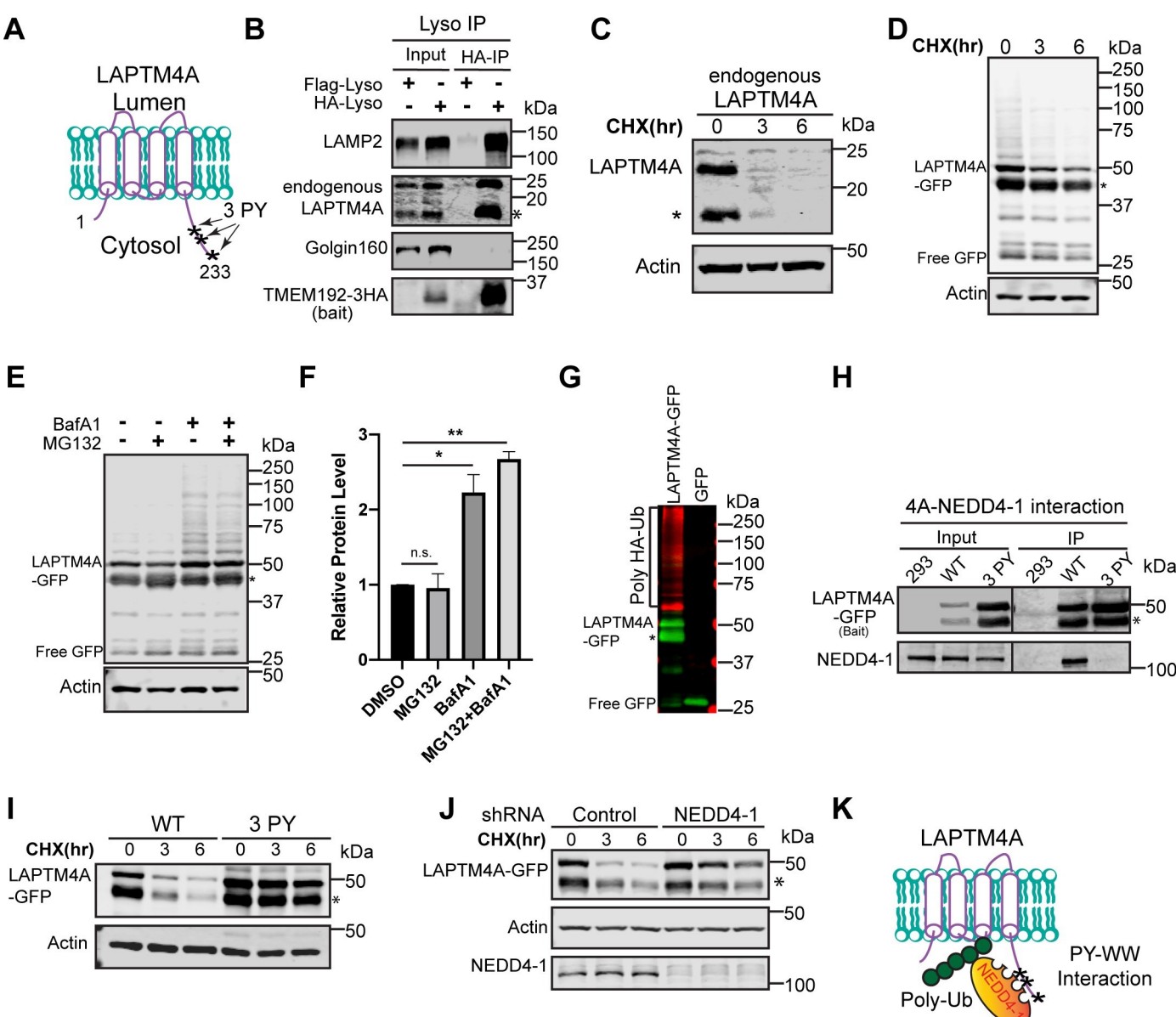

**Fig 6. LAPTM4A degradation depends on NEDD4-1. (A)** A cartoon showing LAPTM4A with 3 PY motifs at its carboxyl terminus. **(B)** Endogenous LAPTM4A is enriched by Lyso-IP. Flag-Lyso: cells stably expressing TMEM192-2XFLAG; HA-Lyso: cells stable expressing TMEM192-3HA. **(C)** Endogenous LAPTM4A is quickly degraded. The asterisk highlights a protease cleavage product. **(D)** CHX chase assay of stably expressed LAPTM4A-GFP in HEK293 cells. **(E)** LAPTM4A-GFP protein levels after MG132 (50 μM), BafA1 (400 nM), and double treatment for 6 hours. **(F)** Total LAPTM4A-GFP protein levels in E, $n = 3$. Error bars represent standard deviation. n.s., not significant. $^*$ $p \leq 0.05$, $^{**}$ $p \leq 0.01$. **(G)** LAPTM4A-GFP, but not GFP, is polyubiquitinated. **(H)** Co-IP showing a PY motif–dependent interaction between LAPTM4A-GFP and NEDD4-1. **(I)** CHX chase assay of stably expressed WT LAPTM4A-GFP and 3PY mutant. **(J)** CHX chase assay of stably expressed LAPTM4A-GFP in shRNA control and NEDD4-1 knockdown cells. **(K)** A cartoon showing the association of LAPTM4A with NEDD4-1 leads to its polyubiquitination. Uncropped western blots are provided in S1 Raw Images. CHX, cycloheximide; shRNA, short hairpin RNA; WT, wild-type.

of *PEP4*, the master vacuolar protease critical for activating other lumenal proteases [55,56], stopped the degradation (Fig 8A and 8B).

We then systematically deleted components of the ESCRT subcomplexes, including *VPS27* (ESCRT-0), *VPS23* (ESCRT-I), *VPS22* (ESCRT-II), *SNF7* (ESCRT-III), *VPS4* (AAA-ATPase), and *BRO1* (ALIX homolog). As shown in Fig 8C, deleting any ESCRT components stabilized

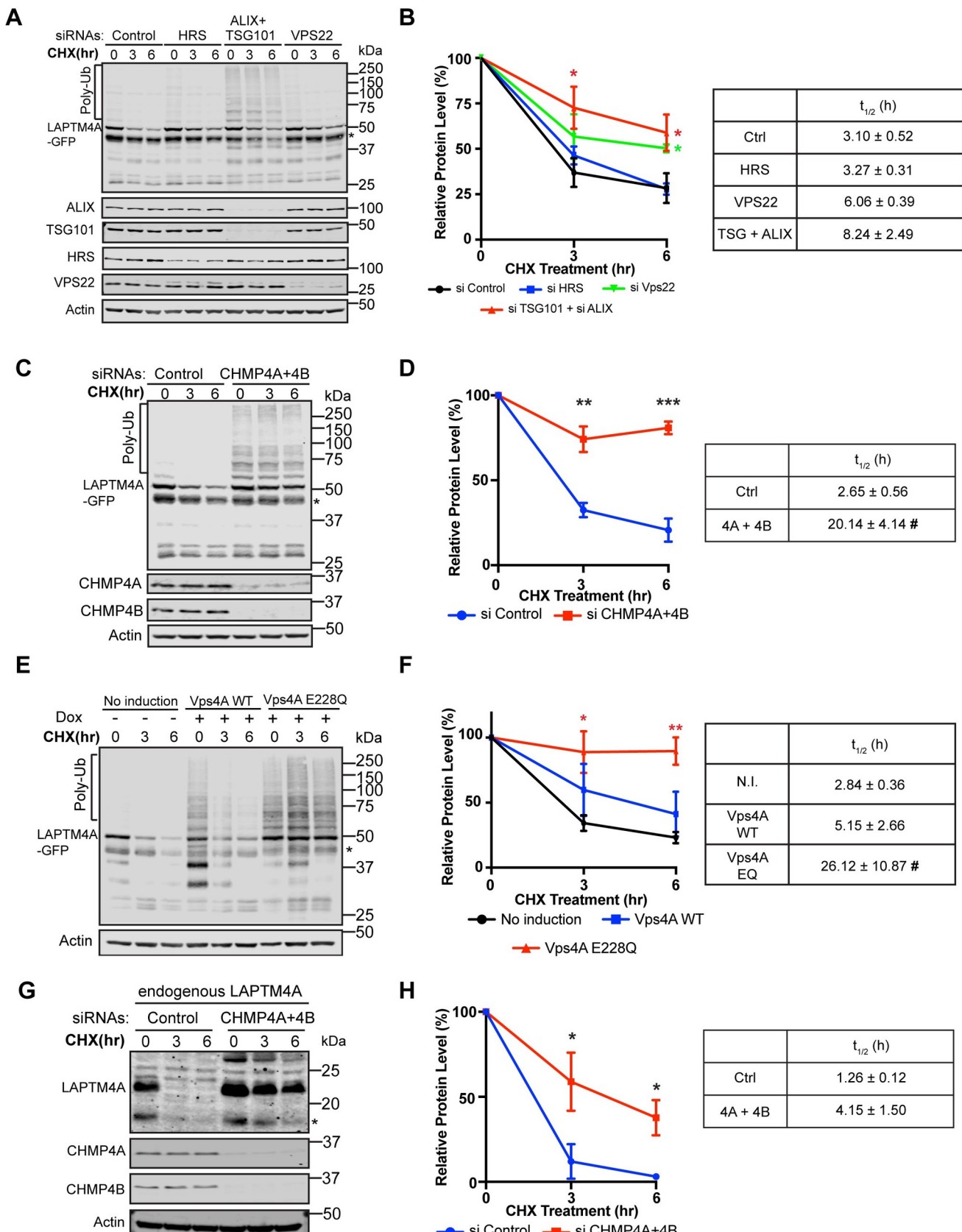

**Fig 7. LAPTM4A degradation is ESCRT dependent. (A)** CHX chase assay of stably expressed LAPTM4A-GFP in siRNA control, HRS knockdown, TSG101 + ALIX double knockdown, and VPS22 knockdown cells. **(B)** Left: total LAPTM4A-GFP protein levels in A, $n$ = 3. Error bars represent standard deviation. * $p \leq 0.05$, ** $p \leq 0.01$, *** $p \leq 0.001$. Right: half-lives of LAPTM4A-GFP in A. **(C)** CHX chase assay of stably expressed LAPTM4A-GFP in siRNA control and CHMP4A + CHMP4B double knockdown cells. **(D)** Left: total LAPTM4A-GFP protein levels in C, $n$ = 3. Right: half-lives of LAPTM4A-GFP in C. #: one of the degradation data sets is too stable to fit with the first-order decay. The average $t_{1/2}$ is calculated based on the other 2 data sets. **(E)** CHX chase assay of stably expressed LAPTM4A-GFP in cells overexpressing either WT HA-VPS4A or its dominant-negative mutant (E228Q). **(F)** Left: total LAPTM4A-GFP protein levels in E, $n$ = 3. Right: half-lives of LAPTM4A-GFP in E. **(G)** CHX chase assay of endogenous LAPTM4A in siRNA control and CHMP4A + CHMP4B double knockdown cells. **(H)** Left: endogenous LAPTM4A protein levels in G, $n$ = 3. Right: half-lives of the endogenous LAPTM4A in G. Uncropped western blots are provided in S1 Raw Images. The numerical data underlying the plot can be found in S1 Data. CHX, cycloheximide; ESCRT, endosomal sorting complexes required for transport; siRNA, small interfering RNA; WT, wild-type.

GFP-RNF152 on the vacuole membrane. These results indicated that the ESCRT machinery is critical for the internalization and degradation of GFP-RNF152 in yeast. Unlike human cells, early-stage ESCRTs are just as crucial as the late-stage ESCRTs in yeast, suggesting that humans may have evolved an alternative mechanism to recruit late-stage ESCRTs onto the lysosome.

In summary, our investigation indicated that the ubiquitin- and ESCRT-dependent LMP turnover is a conserved mechanism from yeast to humans (Fig 8D).

## Discussion

### Different ESCRT requirements between yeast and human lysosomes

This study uncovered a ubiquitin and ESCRT- dependent pathway to down-regulate human LMPs such as RNF152 and LAPTM4A. Strikingly, when expressed in budding yeast, RNF152 is also degraded by a similar mechanism (Fig 8). Previously, we and others have demonstrated that yeast uses the same pathway to regulate vacuole membrane proteins [53,57–60]. For example, the Ssh4-Rsp5 E3 ligase complex ubiquitinates Ypq1 (a vacuolar lysine transporter) when lysine is depleted from the media [53,61]. After ubiquitination, vacuole membrane proteins are internalized by the ESCRT machinery for degradation [53,57,59,60,62,63].

Although the pathway is conserved, we noticed significant differences between yeast and human. In budding yeast, knocking out every subcomplex of the ESCRT machinery (ESCRT-0, I, II, III, and Vps4) stabilizes RNF152 on the membrane (Fig 8). But in humans, early ESCRTs seem to be less important (Fig 7A and 7B, S4 Fig). Only ESCRT-III (CHMP4A and CHMP4B) and VPS4 are essential for the degradation (Figs 5 and 7C–7F). Our results are consistent with a previous study showing that starvation-induced turnover of autophagy receptors by endosomal microautophagy depends on ESCRT-III and VPS4, but not early ESCRTs (ESCRT-0, ESCRT-I, and ESCRT-II) [64].

One possible explanation could be the functional redundancy among early components. Interestingly, many early ESCRTs share functional domains or binding features. For example, both HRS (ESCRT-0) and VPS28 (ESCRT-II) contain PI3P binding domains [65]. Also, many ESCRT components, including HRS, STAM1&2 (ESCRT-0), TSG101, MVB12A&B (ESCRT-I), Vps36 (ESCRT-II), and ALIX, are all ubiquitin-binding proteins [66]. Therefore, it is possible that some early ESCRT components are functionally redundant.

Alternatively, it is also possible that human lysosomes may have evolved other ways to recruit the late-stage ESCRTs. Besides the classic role of ILV formation, ESCRTs have evolved a variety of functions on different membranes, such as plasma membrane repair, budding of viral particles, midbody formation during cytokinesis, and closure of the autophagosome. All these functions require ESCRT-III and VPS4, which are directly responsible for membrane deformation [44]. However, the corresponding proteins to recruit ESCRT-III at different membranes are different. For example, ESCRT-0, CEP55, and viral Gag proteins can all recruit

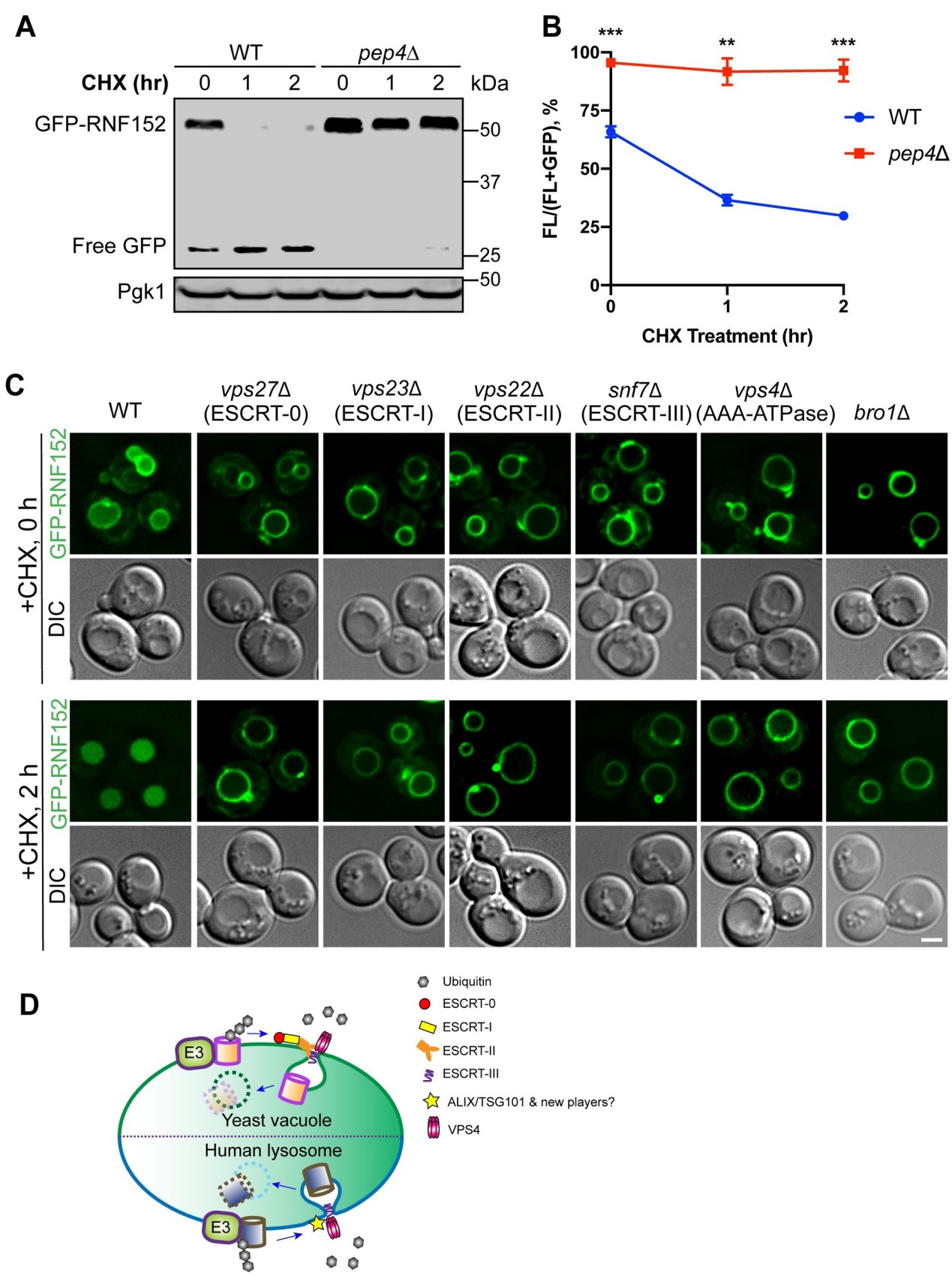

**Fig 8. The degradation of RNF152 is ESCRT dependent in budding yeast.** **(A)** CHX chase assay of GFP-RNF152 in WT and *pep4Δ* yeast strains. **(B)** GFP-RNF152 protein levels in A, *n* = 3. Error bars represent standard deviation. ** $p \leq 0.01$, *** $p \leq 0.001$. **(C)** Subcellular localization of GFP-RNF152 in WT, *vps27Δ*, *vps23Δ*, *vps22Δ*, *snf7Δ*, *vps4Δ*, and *bro1Δ* yeast strains before (0 hour) and after (2 hours) CHX treatment. Scale bar = 2 μm. **(D)** A model showing the conserved ubiquitin- and ESCRT-dependent mechanism to degrade LMPs in yeast and humans. Uncropped western blots and microscopy images are provided in S1 Raw Images. The numerical data underlying the plot can be found in S1 Data. CHX, cycloheximide; DIC, differential interference contrast; WT, wild-type.

ESCRT-III to distinct membrane surfaces [44]. Identifying the new lysosome recruitment factors will require further investigation.

## Multiple pathways may be involved in the selective turnover of LMPs

Besides the ESCRT-dependent pathway uncovered by this paper, recent studies have suggested other mechanisms that might be involved in LMP degradation. First, Lee and colleagues showed that LC3 lipidation–triggered microautophagy is responsible for the selective turnover of several LMPs, including TRPML1 and SNAT7 (SLC38A7) [32]. However, there are still many unanswered questions about this pathway. For example, how does ATG5-dependent microautophagy achieve its selectivity? It is unclear what machinery is responsible for selecting cargo proteins and how selected LMPs are sorted into microautophagy. Despite all these questions, ATG5-dependent microautophagy provides an exciting model for the selective turnover of some LMPs. Although our study did not find evidence that this microautophagy is responsible for the degradation of RNF152, it is still possible that these 2 pathways may operate in parallel to regulate different subsets of LMPs.

The proteasome-dependent degradation is another well-characterized and conserved pathway to down-regulate membrane proteins. The best-known example is the endoplasmic reticulum–associated degradation (ERAD) pathway, in which misfolded proteins were ubiquitinated by the E3 ligases at the endoplasmic reticulum, extracted by the AAA ATPase P97 (CDC48 in yeast), and sent to the proteasome for degradation [67]. Interestingly, several recent studies showed that p97 could be recruited to the lysosome to facilitate lysophagy [68,69]. It is possible that some LMPs extracted by p97 can be delivered to the proteasome for degradation [70]. As we are still at the early stage of understanding lysosome membrane degradation, how ESCRTs, autophagy, and proteasome may coordinate to regulate its membrane composition remains an exciting puzzle.

## Materials and methods

### Mammalian cell culture

All mammalian cell lines used in this study are listed in S1A Table. The following cell lines were obtained from ATCC: HEK293 (CRL-1573), HEK293T (CRL-3216), and HeLa (CCL-2). Cells were cultured in Dulbecco's Modified Eagle Medium (DMEM; Invitrogen, Grand Island, New York, USA.) containing 10% Super Calf Serum (Gemini Bio, West Sacramento, California, USA), 1% penicillin and streptomycin (Invitrogen), and 1 μg/ml plasmocin (InvivoGen, San Diego, California, USA) at 37°C, 5% $CO_2$. All cells were tested negative for mycoplasma contamination using MycoAlert Mycoplasma Detection Kit (Lonza, Houston, Texas, USA).

### Plasmids

All mammalian plasmids used in this study are listed in S1B Table. Most of the LMP overexpression plasmids were purchased from GeneCopoeia Rockville, Maryland, USA. The CDS of RNF152 was purchased from Horizon Discovery Lafayette, Colorado, USA. The coding DNA sequence (CDS) of LAPTM4A was purchased from GeneCopoeia.The 4C➡S and QK➡AA

mutants of RNF152 and 3 PY mutant of LAPTM4A were generated by PCR-based site-directed mutagenesis. The 8K➔R mutant of RNF152 was synthesized as a gBlock gene fragment by Integrated DNA Technologies (Coralville, Iowa, USA).

To generate transfer plasmids for GFP-RNF152 stable cell lines, EGFP was fused to the N-terminal of RNF152 (WT or mutants) using PCR-based overlapping extension, with 2X Gly-Gly-Gly-Ser linker in between. Then, EGFP-RNF152 was cloned into the pHAGE2-IRES-puro vector using restriction enzymes NotI and BamHI. For LAPTM4A-GFP (WT or mutants), GFP was fused to the carboxyl terminus of the LAPTM4A and then cloned into pHAGE2-IRES-puro vector. To generate mCherry selection transfer plasmids, mCherry was amplified from the pmCherry-N1 (Clontech Mountain View, California, USA.) vector and cloned into a pHAGE2 vector to replace the puromycin-resistant gene using restriction enzymes NdeI and ClaI. To generate transfer plasmids for 3XFLAG-RNF152 stable cell lines, RNF152 (WT or mutants) was cloned into the pBICEP-CMV2-3XFLAG (Millipore-Sigma St. Louis, Missouri, USA.) vector using restriction enzymes SalI and BamHI. Then, 3XFLAG-RNF152 was cloned into pHAGE2-IRES-Puro vector using restriction enzymes NcoI and BamHI.

The CDS of Vps4A E228Q was purchased from Addgene Watertown, Massachusetts, USA. (80351). Vps4A E228Q was cloned into the pCMV-HA (Clontech) vector using restriction enzymes EcoRI and NotI. The WT HA-Vps4A was obtained by using PCR-based site-directed mutagenesis. Then the WT HA-Vps4A and HA-Vps4A E228Q were cloned into pCW57.1 (Addgene 41393) using restriction enzymes NheI and AgeI.

## Transfection and cycloheximide chase screen

HEK293 cells were cultured in DMEM containing 10% serum-only media for at least 4 days before transfection. Cells were transfected with individual overexpression plasmids (2.4 μg DNA for a 3.5-cm dish) using Lipofectamine 2000 (Invitrogen) according to the manufacturer's instruction. After 20 to 24 hours of transfection, cells were treated with 100 μg/ml CHX (Millipore-Sigma). At indicated chase time, cell samples were collected in ice-cold 1X PBS, pelleted at 800 x g for 1 minute, and stored at −80˚C before subsequent western blot analysis.

## Generation of lentiviral stable cell lines and shRNA knockdown

Stable cell lines were generated as described in Abu-Remaileh and colleagues with some modifications [26]. HEK293T cells were transfected with transfer plasmid, psPAX2 (Addgene 12260), and pMD2.G (Addgene 12259) at 3.5:3.5:1 ratio using Lipofectamine 2000 according to the manufacturer's instruction. Three days after transfection, the virus-containing supernatant was collected using a 5-ml syringe and applied through a 0.45-μm filter. To generate stable cell lines, HEK293, HEK293T, or HeLa cells were seeded in 3.5-cm dishes and infected with the infectious media (DMEM containing 10% super calf serum, 10 μg/ml polybrene, and multiplicity of infection [MOI] between 0.3 and 0.4). For puromycin selection, the media was refreshed with DMEM containing 10% super calf serum and 1 μg/ml puromycin. The selection lasted for at least 7 days before subsequent analysis. For mCherry selection cells (pHAGE2-GFP-RNF152-IRES-mCherry and pHAGE2-LAPTM4A-GFP-IRES-mCherry), the media was refreshed with DMEM containing 10% super calf serum. Three days after transduction, cells with proportional GFP versus mCherry expression were enriched by fluorescence-activated cell sorting (FACS). Sorting was repeated 7 days after the first round of FACS.

For short hairpin RNA (shRNA) knockdown, the shRNA was cloned into pLKO.1 (Addgene 8453). HEK293 cells were infected with infectious media DMEM containing 10% super calf serum and 10 μg/ml polybrene. After 72 hours, cells were subjected to subsequent CHX chase and western blotting analysis.

The following shRNA sequence for Nedd4-1 knockdown was used in this study: NEDD4-1: 5′-CCGG<u>GCTGAACTATACGGTTCAAAT</u>CTCGAG<u>ATTTGAACCGTATAGT TCAGC</u> TTTTTG-3′.

## siRNA knockdown

HEK293 cells were cultured in DMEM containing 10% serum only media for at least 4 days before siRNA knockdown. Cells were transfected with siRNAs using Lipofectamine RNAimax (Invitrogen) according to the manufacturer's instruction. After 24 hours, cells were transfected with the same amount of siRNA again. Moreover, 72 hours after the first round of transfection, cells were subjected to subsequent western blotting, immunostaining, or flow cytometry analysis.

The following siRNA sequences were used in this study at indicated concentration: TSG101 (50 nM): 5′-CCUCCAGUCUUAUCUCGUC-dTdT-3′ [18]

ALIX (50 nM): 5′-CCUGGAUAAUGAUGAAGGATT-dTdT-3′ [18]

VPS22 (50 nM): 5′-CUUGCAGAGGCCAAGUAUA-dTdT-3′ [71]

HRS (50 nM): 5′-CGACAAGAACCCACACGU-dTdT-3′ [72]

CHMP4A (100 nM): 5′-GGCACAAACUGACGGGACA-dTdT-3′ [73]

CHMP4B (100 nM): 5′-CGAUAAAGUUGAUGAGUUA-dTdT-3′ [64]

ON-TARGETplus Non-targeting Pool (50 nM, Dharmacon (Lafayette, Colorado, USA)): 5′-UGGUUUACAUGUCGACUAA-3′, 5′-UGGUUUACAUGUUGUGUGA-3′, 5′-UGGU UUACAUGUUUUUCUGA-3′, 5′-UGGUUUACAUGUUUUUCCUA-3′.

## Generation of CRISPR/Cas-9 knockout cell lines

ATG5 and ATG7 knockout HEK293 cells were generated as described [33]. The single-guide RNA (sgRNA) guides for ATG5 and ATG7 CRISPR/Cas-9 knockout were described in An and colleagues: 5′-GATCACAAGCAACTCTGGAT-3′ for ATG5 and 5′-ATCCAAGGCAC TACTAAAAG-3′ for ATG7 [34]. In brief, sgRNA guides were ligated into pspCas9(BB)-2A-Puro (Addgene, 48139) plasmid. HEK293 cells were transfected with CRISPR/Cas-9 knockout plasmids using Lipofectamine 2000 according to the manufacturer's instruction. After 24 hours of transfection, cells were treated with 1 μg/ml puromycin (Invitrogen) for 72 hours. Single cells were isolated into 96-well plates using limited dilution to a final concentration of 0.5 cell per well. The knockout colonies were screened by western blot analysis using antibodies against ATG5 and/or ATG7. The knockout cell lines were further verified by sequencing analysis to confirm the indels at target sites.

## Yeast strains, plasmids, media, and growth conditions

All yeast strains and plasmids used in this study are listed in S1C Table. Difco Yeast Nitrogen Base (YNB) without amino acids were purchased from Millipore-Sigma. Yeast strains were grown in YNB at 26°C before further analysis.

## Mammalian sample preparation and western blotting

Cells were collected in ice-cold 1X PBS, pelleted at 800 x g for 1 minute, and lysed in lysis buffer (20 mM Tris pH = 8.0, 150 mM NaCl, and 1% Triton) containing 1X protease inhibitor cocktail (BioTool, Jupiter, Florida, USA) at 4°C for 20 minutes. Cell lysates were centrifuged at 18,000 g for 15 minutes at 4°C. The protein concentration of the supernatant was measured by Bradford assay (Bio-Rad, Hercules, California, USA.) and normalized. After adding 2X urea sample buffer (150 mM Tris pH 6.8, 6 M Urea, 6% SDS, 40% glycerol, 100 mM DTT, and 0.1% Bromophenol blue), samples were heated at 65°C for 8 minutes. Moreover, 20 μg of each lysate

was loaded and separated on 11% SDS-PAGE gel. Protein samples were transferred to nitrocellulose membrane for western blot analysis. After incubated with primary and secondary antibodies, membranes were scanned using the Odyssey CLx imaging system (LI-COR Biosciences, Lincoln, Nebraska, USA).

The following primary antibodies were used for western blotting in this study: rabbit anti-GFP (1:3,000, TP401, Torrey Pines Biolabs, Secaucus, New Jersey, USA), mouse anti-actin (1:5,000, Proteintech, Rosemont, Illinois, USA), mouse anti-LAMP1 (1:1,000, H4A3, DHSB, University of Iowa, Iowa City, Iowa, USA), mouse anti-LAMP2 (1:1,000, H4B4, DHSB), rabbit anti-CTSD (1:1,000, Cell Signaling Technology, Danvers, Massachusetts, USA), mouse anti-PDI (1:2,000, BD Biosciences, San Jose, California, USA), rabbit anti-Golgin160 (1:1,000, Proteintech), mouse-anti-EEA (1:500, G-4, Santa Cruz Biotechnology, Dallas, Texas, USA), rabbit-GAPDH (1:2,000, Proteintech), mouse anti-ubiquitin (1:100, P4D1, Santa Cruz Biotechnology), rabbit anti-ATG5 (1:2,000, D5FF5U, Cell Signaling Technology), rabbit anti-ATG7 (1:2,000, D12B11, Cell Signaling Technology), rabbit anti-FLAG (1:2,000, Millipore-Sigma), mouse anti-TSG101 (1:200, C-2, Santa Cruz Biotechnology), mouse anti-ALIX (1:500, 1A12, Santa Cruz Biotechnology), mouse anti-Vps22 (1:500, C-11, Santa Cruz Biotechnology), rabbit anti-HRS (1:2,000, D7T5N, Cell Signaling Technology), mouse anti-HA (1:1,000, 16B12, BioLegend, San Diego, California, USA), rabbit anti-CHMP4B (1:2,000, Proteintech), and rabbit anti-LAPTM4A (1:1,000, HPA, Millipore-Sigma).

The rabbit anti-RNF152 antibody was raised against the cytosolic domain of RNF152 (amino acid 1 to 166) expressed in BL21 competent *Escherichia coli*. The antibody was generated in rabbits by Covance (Princeton, New Jersey, USA). The immunoreactive sera were further affinity-purified against immobilized RNF152 (1 to 166). This antibody was used at 1:500 dilution in western blot analysis.

The following secondary antibodies were used in this study: goat anti-mouse IRDye 680LT, goat anti-mouse IRDye 800CW, goat anti-rabbit IRDye 680LT, and goat anti-rabbit IRDye 800CW. All secondary antibodies were purchased from LI-COR Biosciences and used at 1:10,000 dilution.

To detect the endogenous ubiquitin using a anti-Ub (P4D1, Santa Cruz Biotechnology) antibody, antigen retrieval was performed by boiling the membrane in 1XTBS for 30 minutes before blocking.

## Yeast sample preparation and western blotting

Typically, 7 $OD_{600}$ unit yeast cells were collected for sample preparation at each timepoint. The cells were first resuspended with 10% ice-cold TCA and incubated on ice for at least 1 hour. After washing with 0.1% TCA, the cell pellets were then resuspended in 70 μl 2× boiling buffer (150 mM Tris, pH 6.8; 6 M urea; 6% SDS; 10% glycerol; 100 mM DTT), lysed by bead beating for 5 minutes, and heat-treated at 65°C for another 5 minutes. After the addition of 70 μl 2 × urea sample buffer (50 mM Tris, pH 7.5; 6 M urea; 10 mM EDTA; 2% SDS; 100 mM DTT, and 0.1% bromophenol blue), samples were treated for another round of bead beating and 65°C heating for 5 minutes, respectively. After spinning at 13,000 g for 5 minutes, the supernatant was collected, subjected to SDS-PAGE, and transferred to nitrocellulose membranes for western blotting analysis. One $OD_{600}$ unit (20 μl) of yeast cells was loaded in each lane.

The following primary antibodies were used in this study: mouse anti-Pgk1 (1:5,000, 22C5D8; Invitrogen) and rabbit anti-GFP (1:3,000, TP401, Torrey Pines Biolabs).

## Immunostaining for mammalian samples

All incubation processes were performed in the dark. Cells grown on 1.5 circular glass coverslips were washed with ice-cold 1X PBS and fixed in cold 100% methanol for 8 minutes at

−20°C. The fixed samples were blocked in 3% BSA (in 1X PBS) for 30 minutes at room temperature, followed by incubating with primary and secondary antibodies. The cell nucleus was stained using Hoechst (1:8000, Invitrogen). Coverslips were mounted in Fluoromount-G (SouthernBiotech, Birmingham, Alabama, USA) and cured for at least 24 hours before imaging.

The following primary antibodies were used for immunostaining in this study: mouse anti-LAMP2 (1:100, H4B4, DHSB) and mouse anti-ubiquitin (1:100, FK2, Millipore-Sigma).

The following secondary antibodies were used in this study: FITC goat anti-mouse (1:100, Jackson ImmunoResearch, West Grove, Pennsylvania, USA) and TRITC goat anti-mouse (1:100, Jackson ImmunoResearch).

## Microscopy and image processing

Samples were imaged with either a DeltaVision system (GE Healthcare Life Sciences) or a Leica SP8 confocal microscope system [57]. The DeltaVision microscope was equipped with a scientific CMOS camera and an Olympus UPLXAP0100X objective. The filter sets FITC (excitation, 475/28; emission, 525/48), TRITC (excitation 542/27; emission 594/45), and DAPI (excitation 390/18; emission 435/48) were used for GFP, mCherry, and DAPI, respectively. Image acquisition and deconvolution were performed with the softWoRx program.

The Leica SP8 confocal microscope system was equipped with a Lecia PMT (photomultiplier tube) and a HyD (hybrid detector) light-detecting sensor systems. The Leica HC PL APO63X/1.40 Oil CS2 and HC PL APO100X/1.40 Oil STED White objectives were used in this study. Image acquisition was performed with the LAS X Life Science imaging software.

Images were further cropped or adjusted using ImageJ (National Institutes of Health, Bethesda, Maryland, USA).

## Ubiquitin immunoprecipitation with GFP-Trap A nanobody

The HA-Ubiquitin plasmid was transfected into HEK293 cells that either stably express free GFP control, GFP-RNF152, or LAPTM4A-GFP. Immunoprecipitation was performed 48 hours post-transfection according to the manufacturer's instruction with some modifications. In brief, cells (one 10-cm dish of near-confluent cells per IP group) were collected in ice-cold 1X PBS, pelleted at 1,000 g for 1 minute, and lysed in 300 μl of lysis buffer (20 mM Tris pH = 8.0, 150 mM NaCl, 1% Triton) containing 100 mM of N-Ethylmaleimide (Millipore-Sigma) and 1X protease inhibitor cocktail (BioTool) at 4°C for 20 minutes. Cell lysates were centrifuged at 18,000 g for 15 minutes at 4°C. The concentration of the supernatant was measured by Bradford assay (Bio-Rad) and normalized. Moreover, 15 μl of GFP-Trap A (pre-equilibrated with lysis buffer, Chromotek, Islandia, New York, USA) was added to 285 μl of normalized cell lysate and incubated at 4°C for 2 hours with gentle rocking. The resin was then washed once with lysis buffer, 3 times with stringent washing buffer (8M Urea, 1% SDS in 1X PBS), and once with 1% SDS in 1X PBS. To elute bound proteins, the resin was incubated with 2X Urea sample buffer (150 mM Tris pH 6.8, 6 M Urea, 6% SDS, 40% glycerol, 100 mM DTT, and 0.1% Bromophenol blue) at 65°C for 10 minutes. The resulting eluates were analyzed by western blotting.

## Lyso-IP

Lyso-IP was conducted as described before [26] with some modifications. About $2 \times 10^7$ cells in a 15-cm dish were used for each Lyso-IP experiment. Cells were rinsed twice with cold PBS, scraped, and collected with 1-ml KPBS (136 mM KCl, 10 mM $KH_2PO_4$, pH = 7.25). Cells were centrifuged at 1,000 g for 2 minutes at 4°C. Pelleted cells were resuspended in 1 ml of KPBS with protease inhibitor and homogenized. The homogenate was then centrifuged at 1,000 g for 2 minutes at 4°C. For input, 40 μl of supernatant was taken (about 5% to the total amount) and

mixed with 2X Urea sample buffer (150 mM Tris pH 6.8, 6 M Urea, 6% SDS, 40% glycerol, 100 mM DTT, and 0.1% Bromophenol blue). For IP, 800-μl supernatant was incubated with 20-μl anti-HA beads (Millipore-Sigma) and rotated for 20 minutes. The beads were washed with KPBS 6 times. To eluate bound lysosomes, the beads were resuspended in 40-μl KPBS and 2X Urea sample buffer (150 mM Tris pH 6.8, 6 M Urea, 6% SDS, 40% glycerol, 100 mM DTT, and 0.1% Bromophenol blue). Samples were heated at 65˚C for 10 minutes, followed by western blot analysis.

## Flow cytometry analysis and FACS

Cells were washed with 1X PBS and trypsinized until all cells are dissociated from the dishes. Dissociated cells were neutralized with DMEM containing 10% serum media and pelleted at 300 x g for 3 minutes. Cells were resuspended in ice-cold 1X PBS and analyzed using either an LSR Fortessa (BD Biosciences) or a Ze5 (Bio-Rad) flow cytometer. For FACS, cells were sorted using FACSAria III cell sorter (BD Biosciences). Flow cytometry analysis and FACS were performed by technicians from the Flow Cytometry Core at the University of Michigan. The data were analyzed using FlowJo software. An example of Flow Cytometry Standard (FCS) data processing can be found in S2 Data.

## Quantification and statistical analysis

The band intensity for western blot was quantified using Image Studio software (LI-COR Biosciences). To calculate protein half-lives, the data were fitted to the first-order decay, and the rate constant (k) was generated using Excel. The half-lives were calculated using the following equation: $t(_{1/2}) = \ln 2 / k$. Graphs were generated using Prism (GraphPad). Statistical analysis was performed with the 2-tailed unpaired *t* test or 1-way ANOVA. Error bars represent the standard deviation. * $\leq 0.05$, ** $\leq 0.01$, *** $\leq 0.001$.

## Supporting information

**S1 Fig. CHX chase assay to measure half-lives of LMPs. Related to Fig 1.** Some examples of the CHX chase screen in HEK293 cells. LAMP1 and LAMP2 were detected with endogenous antibodies. For other LMPs, HEK293 cells were transiently transfected with indicated overexpression plasmids and detected by either GFP or mCherry antibodies. The majority of the tested LMPs are very stable. Uncropped western blots are provided in S1 Raw Images. CHX, cycloheximide; LAMP, lysosome-associated membrane protein; LMP, lysosomal membrane protein.
(EPS)

**S2 Fig. Endogenous RNF152 is too low to be detected by a homemade antibody. Related to Fig 2. (A)** Endogenous RNF152 (approximately 23 kDa) can only be detected after BafA1 treatment. **(B)** Lanes 2–4: CHX chase assay of RNF152 expressed from a leaky TET-ON promoter. Lane 5: RNF152 after the doxycycline induction, only 10% lysate was loaded. Lanes 6–7: BafA1 treatment stabilized RNF152 from the leaky TET-ON promoter. The white dashed box highlights the position of RNF152. **(C)** Overexpressed GFP-RNF152 can still interact and ubiquitinate its substrate RagA. Uncropped western blots are provided in S1 Raw Images. CHX, cycloheximide.
(EPS)

**S3 Fig. GFP-RNF152 accumulates in the lysosome lumen after BafA1 treatment. Related to Fig 4. (A)** Localization of GFP-RNF152 and LAMP2 after 6 hours BafA1 treatment. The image was taken by a Leica STED microscope. Scale bar: 10 μm. **(B)** Zoomed in pictures of A.

(C) Line scan analysis highlighted the lumenal localization of GFP-RNF152. The uncropped microscopy image is provided in S1 Raw Images. LAMP2, lysosome-associated membrane protein 2; STED, Stimulated Emission Depletion.
(EPS)

**S4 Fig. GFP-RNF152 is mainly degraded by lysosome-dependent proteolysis. Related to Fig 4.** (A) CHX chase assay of stably expressed GFP-RNF152 after MG132 (50 μM), BafA1 (400 nM), and double treatment. (B) Left: full-length GFP-RNF152 levels in A, $n = 3$. Error bars represent standard deviation. $^* p \leq 0.05$, $^{**} p \leq 0.01$. Right: half-lives of GFP-RNF152 in A. Uncropped western blots are provided in S1 Raw Images. The numerical data underlying the plot can be found in S1 Data. CHX, cycloheximide.
(EPS)

**S5 Fig. Macroautophagy and CMA pathways are not involved in the degradation of RNF152.** (A) CHX chase assay of stably expressed GFP-RNF152 in WT, ATG5KO, and ATG7KO cell lines. (B) Left: full-length GFP-RNF152 protein levels in A, $n = 3$. Error bars represent standard deviation. Right: half-lives of GFP-RNF152 in A. (C) western blots showing stably expressed GFP-RNF152 protein levels in ATG7KO cells after indicated treatment for 4 hours. (D) GFP-RNF152 protein levels in C, $n = 3$. $^{**} p \leq 0.01$. (E) Identifying KFERQ-like CMA motif in RNF152. (F) CHX chase assay of WT 3XFLAG-RNF152 and QK→AA mutant. (G) Left: quantification of F, $n = 3$. Right: half-lives of 3xFLAG-RNF152 in F. Uncropped western blots are provided in S1 Raw Images. The numerical data underlying the plot can be found in S1 Data. CHX, cycloheximide; CMA, chaperone-mediated autophagy; WT, wild-type.
(EPS)

**S6 Fig. Early ESCRT components are less important for the degradation of RNF152. Related to Fig 5.** (A) CHX chase assay of stably expressed GFP-RNF152 in indicated knockdown cells. (B) Left: total GFP-RNF152 protein levels in A, $n = 3$. Right: half-lives of GFP-RNF152 in A. Error bars represent standard deviation. (C) CHX chase assay of stably expressed GFP-RNF152 in siRNA control, HRS knockdown, and Vps22 knockdown cells. (D) Left: total GFP-RNF152 protein levels in C, $n = 3$. Right: half-lives of GFP-RNF152 in C. (E) FK2 antibody staining of polyubiquitinated proteins in indicated knockdown cells. Nuclei were labeled with Hoechst. Scale bar = 10 μm. Uncropped western blots and microscopy images are provided in S1 Raw Images. The numerical data underlying the plot can be found in S1 Data. CHX, cycloheximide; ESCRT, endosomal sorting complexes required for transport.
(EPS)

**S7 Fig. Cells overexpressing VPS4A EQ mutant accumulate polyubiquitinated proteins. Related to Fig 5.** FK2 antibody staining of polyubiquitinated proteins in indicated cells. The overexpression of HA-VPS4A was induced by 1 μg/ml of doxycycline treatment for 24 hours. Nuclei were labeled with Hoechst. Scale bar = 10 μm. Uncropped microscopy images are provided in S1 Raw Images.
(EPS)

**S1 Data. Numerical raw data.** This Excel file contains multiple sheets, each of which contains the data that were used to generate plots in Figs 1 and 2–8 and S3–S6 Figs (separated into different sheets for each figure panel).
(XLSX)

**S2 Data. An example of FCS files processing.** An example of the gating applied to FCS files in Fig 2D. FCS, Flow Cytometry Standard.
(PDF)

**S1 Raw Images. Uncropped western blots and imaging data used in all figures.**
(PDF)

**S1 Table. List of mammalian cell lines, yeast strains, and plasmids used in this study.**
(DOCX)

## Acknowledgments

We thank the Li laboratory members, including F. Arines, G. Shi, A. Hamlin, and J. Zhang
from the Wang laboratory, for their technical support.

## Author Contributions

**Conceptualization:** Weichao Zhang, Ming Li.

**Data curation:** Weichao Zhang, Ming Li.

**Formal analysis:** Weichao Zhang, Xi Yang.

**Funding acquisition:** Ming Li.

**Investigation:** Weichao Zhang, Xi Yang, Liang Chen, Yun-Yu Liu, Varsha Venkatarangan,
Lucas Reist.

**Methodology:** Weichao Zhang, Liang Chen.

**Project administration:** Ming Li.

**Resources:** Phyllis Hanson, Haoxing Xu, Yanzhuang Wang, Ming Li.

**Supervision:** Ming Li.

**Validation:** Weichao Zhang, Xi Yang, Yun-Yu Liu.

**Visualization:** Weichao Zhang.

**Writing – original draft:** Weichao Zhang.

**Writing – review & editing:** Ming Li.

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
