## [Editor Report · Decision Letter 0]

18 May 2021

Dear Dr Li, 

Thank you for submitting the revision of your manuscript entitled "A conserved ubiquitin- and ESCRT-dependent pathway to regulate human lysosomal membrane proteins" for consideration as a Research Article by PLOS Biology in the Review Commons program.

Your manuscript and the reviews have now been evaluated by the PLOS Biology editorial staff as well as by an academic editor with relevant expertise and I am writing to let you know that we would like to send your submission back to the original reviewers.

However, before we can do that, we need you to complete your submission by providing the metadata that is required for all our manuscripts. To this end, please login to Editorial Manager where you will find the paper in the 'Submissions Needing Revisions' folder on your homepage. Please click 'Revise Submission' from the Action Links and complete all additional questions in the submission questionnaire.

Please re-submit your manuscript within two working days, i.e. by May 20 2021 11:59PM.

Kind regards,

Ines

--

Ines Alvarez-Garcia, PhD

Senior Editor

PLOS Biology

---

## [Decision Letter · Decision Letter 1]

15 Jun 2021

Dear Dr Li,

Thank you for submitting your revised Research Article entitled "A conserved ubiquitin- and ESCRT-dependent pathway to regulate human lysosomal membrane proteins" for publication in PLOS Biology. I have now obtained advice from two of the original reviewers from Review Commons and have discussed their comments with the Academic Editor. 

Based on the reviews (attached below), we will probably accept this manuscript for publication, provided you satisfactorily address the remaining points raised by Reviewer 3. Please also make sure to address all the data and policy-related requests included below. In addition, we would like you to consider a suggestion to improve the title to:

"A conserved ubiquitin- and ESCRT-dependent pathway internalizes human lysosomal membrane proteins for degradation"

We expect to receive your revised manuscript within two weeks. 

-  a Response to Reviewers file that provides a detailed response to the reviewers' comments

*Published Peer Review History*

*Early Version*

Sincerely,

Ines

--

Ines Alvarez-Garcia, PhD,

Senior Editor,

ialvarez-garcia@plos.org,

PLOS Biology

Figs. 1B; Fig. 2D; Fig. 3D, E; Fig. 4B, C, E, G; Fig. 5B, C, E, G, H; Fig. 6F; Fig. 7B, D, F, H; Fig. 8B; Fig. S2B; Fig. S3B, D, G and Fig. S4B, D

**For figures containing FACS data, we ask that you provide FCS files and a picture showing the successive plots and gates that were applied to the FCS files. As the data is quite extensive, we can suggest to deposit it at the Flow Repository (https://flowrepository.org/). If you do this, please provide the ID number in the Data Availability Statement and make sure it is made publicly available at this stage.

We require the original, uncropped and minimally adjusted images supporting all blot and gel results reported in an article's figures or Supporting Information files. We will require these files before a manuscript can be accepted so please prepare and upload them now. Please carefully read our guidelines for how to prepare and upload this data: https://journals.plos.org/plosbiology/s/figures#loc-blot-and-gel-reporting-requirements 

Reviewers' comments

Rev. 2:

The authors have taken the effort to address all my previous comments. I do not have any further concerns. Nice work! 

Rev. 3:

The authors have revealed details concerning how two lysosome membrane proteins, RNF152 and LAPTM4A, are internalized into lysosomes for degradation in a ubiquitin and ESCRT dependent manner. Although it remains uncertain to what extent these examples can be generalized to other lysosome membrane proteins, these are interesting new findings. Nonetheless, given that the authors have only characterized 2 proteins that follow this degradation pathway, caution is required with respect to generalization. The authors were responsive to reviewer comments. My only remaining concerns are minor and should not require additional experiments.

1. Data presented in the response to reviewers should be included in the manuscript. If the reviewers need this information to assess the results, then other readers will appreciate them as well. That said, the preliminary screening data need not be added.

2. It remains unclear whether the RNF152 degradation observed with the over-expressed protein reflects the normal degradation pathway for the endogenous protein. Either way, it could be a useful tool. However, the authors should be more up front with this limitation if they cannot show that the endogenous protein is degraded in a similar manner.

3. The authors state: "Without treatment,many LSD patients will develop severe neurodegeneration symptoms (Marques and Saftig, 2019)." This sentence in the introduction is problematic as it implies that there are treatments for the neurological aspects of lysosome storage diseases. Unfortunately, treatments such as enzyme replacement therapy are only effective for the non-neuronal symptoms.

4. Page 27: Gene transfer with replication incompetent lentivirus is referred to as transduction rather than infection.

---

## [Editor Report · Decision Letter 2]

9 Jul 2021

Dear Dr Li,

On behalf of my colleagues and the Academic Editor, Ana Garcia-Saez, I am pleased to say that we can in principle offer to publish your Research Article entitled "A conserved ubiquitin- and ESCRT-dependent pathway internalizes human lysosomal membrane proteins for degradation" in PLOS Biology, provided you address any remaining formatting and reporting issues. These will be detailed in an email that will follow this letter and that you will usually receive within 2-3 business days, during which time no action is required from you. Please note that we will not be able to formally accept your manuscript and schedule it for publication until you have made the required changes.

PRESS

Sincerely, 

Ines

--

Ines Alvarez-Garcia, PhD 

Senior Editor 

PLOS Biology
